# COVID-19 in Pakistan: A national analysis of five pandemic waves

**Taimoor Ahmad**[1], **Mujahid Abdullah**[1], **Abdul Mueed**[1], **Faisal Sultan**[2,3], **Ayesha Khan**[4], **Adnan Ahmad Khan**[1,2]*

1 Research and Development Solutions, Islamabad, Pakistan, 2 Ministry of National Health Services, Regulation and Coordination, Islamabad, Pakistan, 3 Shaukat Khanum Memorial Cancer Hospital & Research Centre, Lahore, Pakistan, 4 Akhter Hameed Khan Foundation, Islamabad, Pakistan

* adnan@resdev.org

**Data Availability Statement:** The data was provided to the Akhter Hameed Khan Foundation (AHK-F) team for this study as part of its work with Pakistan's Federal Ministry of National Health Services, Regulations & Coordination

## Abstract

### Objectives

The COVID-19 pandemic showed distinct waves where cases ebbed and flowed. While each country had slight, nuanced differences, lessons from each wave with country-specific details provides important lessons for prevention, understanding medical outcomes and the role of vaccines. This paper compares key characteristics from the five different COVID-19 waves in Pakistan.

### Methods

Data was sourced from daily national situation reports (Sitreps) prepared by the National Emergency Operations Centre (NEOC) in Islamabad. We use specific criteria to define COVID-19 waves. The start of each COVID-19 wave is marked by the day of the lowest number of daily cases preceding a sustained increase, while the end is the day with the lowest number of cases following a 7-days decline, which should be lower than the 7 days following it. Key variables such as COVID-19 tests, cases, and deaths with their rates of change to the peak and then to the trough are used to draw descriptive comparisons. Additionally, a linear regression model estimates daily new COVID-19 deaths in Pakistan.

### Results

Pakistan saw five distinct waves, each of which displayed the typical topology of a complete infectious disease epidemic. The time from wave-start to peak became progressively shorter, and from wave-peak to trough, progressively longer. Each wave appears to also be getting shorter, except for wave 4, which lasted longer than wave 3. A one percent increase in vaccinations decreased deaths by 0.38% (95% CI: -0.67, -0.08) in wave 5 and the association is statistically significant.

### Conclusion

Each wave displayed distinct characteristics that must be interpreted in the context of the level of response and the variant driving the epidemic. Key indicators suggest that COVID-19 preventive measures kept pace with the disease. Waves 1 and 2 were mainly about

(MoNHSR&C) and the National Command & Operation Centre (NCOC) in Islamabad, which are leading Pakistan's response to the COVID-19 pandemic. The AHK-F team has provided analytical support to the above entities, and such created knowledge that has directly informed pandemic policy-making in Pakistan. COVID-19 data is compiled and shared in daily National Situation Reports, or Sitreps, by the National Emergency Operation Centre (NEOC). Each day's Sitrep is compiled as a PDF file. The data used for this study was manually compiled from these PDF files and then used in STATA. The parentage of this data is with the NCOC and the MoNHSR&C. The AHK-F team received this data with the express understanding that it would be kept confidential. However, the data can be obtained independently from the NEOC, through a data request procedure, which is subject to approval from the MoNHSR&C. The data request itself is to be addressed to: Dr. Shahzad Baig, National Coordinator, National Emergency & Operation Center, D Block, EPI Building, Chak Shahzad, Park Road, Islamabad. Email: eocpakistan@gmail.com Phone: +92-51-8730879. The data on Oxford Health and Containment Index is taken from and publicly available at the following GitHub repository: https://github.com/OxCGRT/covid-policy-tracker/tree/master/data.

**Funding:** This work was supported, in whole or in part, by the Bill & Melinda Gates Foundation [grant number: INV-025171]. Under the grant conditions of the Foundation, a Creative Commons Attribution 4.0 Generic License has already been assigned to the Author Accepted Manuscript version that might arise from this submission. The funders had no role in study design, data collection and analysis, decision to publish, or preparation of the manuscript.

**Competing interests:** The authors have declared that no competing interests exist.

prevention and learning how to clinically manage patients. Vaccination started late during wave 3 and its impact on hospitalizations and deaths became visible in wave 5. The impact of highly virulent strains Alpha/B.1.1.7 and Delta/B.1.617.2 variants during wave 3 and milder but more infectious Omicron/B.1.1.529 during wave 5 are apparent.

## Introduction

In Pakistan, the first case of COVID-19, a novel and little-understood disease, was detected on February 26, 2020. Being a developing country with limited resources, crumbling health infrastructure and low health expenditure [1], Pakistan has no past experience with pandemics and a high burden of communicable diseases [2]. As of February 23, 2022, the country had fully vaccinated 43% of its total population [3] and the Omicron variant of COVID-19 was the dominant strain [4]. These factors make Pakistan a high-risk country, with a large pool of infection-susceptible people.

The emergence of COVID-19 has arguably been the biggest social and economic disruption in Pakistan in recent history. The pandemic has largely manifested itself in five distinct waves each of which have a rise, plateau, and trough in cases, followed by a period of dormancy, after which the incidence of COVID-19 infections begins to rise again. Thus, each individual wave follows a four-stage pattern followed by endemicity that has been seen for many infectious disease epidemics [5]. Beyond anecdotal observation, there is evidence that this is happening with COVID-19 as well [6]. What sets COVID-19 apart is that after completion of an individual wave, a new one would come along shortly, rather than taking much long, for example, annual recurrences for influenza. This pattern has been seen across the globe [7–11] with the timing of COVID-19 waves in different countries broadly coinciding [6].

In this context, current literature on COVID-19 largely focuses on high-income countries during the initial waves [7,12,13], or aggregated at regional levels [8,14]. Given the different capacities of countries to manage the pandemic [15], there is a need to explore the characteristics of the subsequent pandemic waves in a developing country context, preferably with granularity of a country-level analysis.

This paper aims to offer a comprehensive understanding of the impact of COVID-19 in Pakistan. To achieve this, we examine the five waves of the pandemic in Pakistan, analyzing various key aspects and critical statistics. These include the total number of COVID-19 tests conducted, confirmed cases, hospitalizations, COVID-19-related deaths, and the progress of vaccinations during each wave. Additionally, we employ statistical modeling to identify the significant factors contributing to COVID-19-related deaths. Our goal is to fill the existing gap in the literature by providing valuable insights specific to a developing country like Pakistan, where limited evidence currently exists.

## Methods

### Criteria for COVID-19 waves

We begin by retrospectively defining various time periods between 2020 and 2022 as distinct waves, based on existing literature [16]. There are a total of 628 observations (daily set of indicators) across these five waves. Based on our criteria, the starting point of each COVID-19 wave is defined as the day with the lowest number of daily new COVID-19 cases preceding a consistent rise in these cases, before the peak of the respective COVID-19 waves. The end of

each wave is defined as the day with lowest number of daily new COVID-19 cases following a 7-day decline; this number also needed to be lower than the cases on any of the 7 days that followed it (Table 2).

## Data and variables

In order to estimate the pattern for COVID-19 throughout the five waves in Pakistan, we use time series data of various daily indicators from April 3, 2020 to February 23, 2022, which are categorized into the following broad themes:

i) Wave timespan

ii) COVID-19 tests

iii) COVID-19 cases

iv) Test-to-case ratio

v) COVID-19 positivity

vi) Hospitalization and treatment

vii) COVID-19 deaths

viii) COVID-19 vaccination

ix) Policy environment

Several variables in the list above were transformed into ratios for the purpose of describing all five COVID-19 waves in Pakistan (S1 Table).

The data for all but two of the above themes, COVID-19 vaccination and policy environment, is compiled from daily national situation reports (Sitreps). These Sitreps are prepared by the National Emergency Operations Centre (NEOC) in Islamabad, Pakistan. Data in these Sitreps have served as the basis for all major COVID-19 policy decisions in Pakistan.

The data for COVID-19 vaccination is sourced directly from the National Command & Operation Centre (NCOC), Islamabad, Pakistan, which is the government forum that brings together the ministries of Health and Planning along with the military to determine pandemic policy and to coordinate the response. Data for the policy environment is taken from a publicly available dataset from the University of Oxford's Blavatnik School of Governance [17]. This dataset is compiled by using qualitative information about the non-pharmaceutical interventions (NPIs) in a country and quantifying them into an index called Oxford Containment and Health Index for COVID-19. A detailed methodology of the index calculation can be found in a working paper by the Blavatnik school [18].

## Model specification

Apart from presenting statistics on daily indicators for every wave, we estimate the predictors of daily new deaths due to COVID-19. For our model of daily new COVID-19 deaths, we use a linear ordinary least square (OLS) regression. The data as well as the model is divided into five distinct periods, representing the five waves of COVID-19 in Pakistan, as of February 2022. The manuscript comprises statistical analysis and inferences for each of the five waves separately.

Our dependent variable is the daily new COVID-19 deaths in logarithmic form ($\mathbf{LnY_t}$); $\epsilon_t$ represents the error term capturing the effect of omitted variables. The specification of our

linear OLS regression model is as follows:

$$Ln\,Y_t = \beta_0 + \beta_1\,Ln\,X1_{t+21} + \beta_2\,Ln\,X2_{t+28} + \beta_3\,X3_{t+14} + \beta_4\,X4_t + \beta_5\,X5_t + \beta_6\,X6_t + \beta_7\,Ln\,X7_{t+14} + \epsilon_t \quad (1)$$

Our independent variables measured at daily intervals are:

i) Log of daily new COVID-19 cases with 21-day delay ($LnX1_{t+21}$);

ii) Log of daily new COVID-19 tests with 28-day delay ($LnX2_{t+28}$);

iii) The Oxford containment and health index for COVID-19 with 14-day delay ($X3_{t+14}$);

iv) Time variable capturing the time trend ($X4_t$);

v) The number of people on ventilators as a proportion of the total admitted ($X5_t$);

vi) The number of people on oxygen as a portion of the total admitted ($X6_t$);

vii) Log of second doses of COVID-19 vaccines administered with 14-day delay ($LnX7_{t+14}$)

Daily new COVID-19 cases are regressed with a 21-day lag, since among those who die from COVID-19 infection, death occurs between a median of 14 days [19] and 25 days (average of three weeks) after presenting symptoms [20,21]. This is pertinent in the case of Pakistan, as most of the COVID-19 testing in the country has been symptomatic, i.e., done when someone develops symptoms of COVID-19 and hence either voluntarily gets tested or is prescribed by a medical professional to do so.

Given the delay for daily new cases, daily new COVID-19 tests are regressed with a delay of 28 days. This delay allows for the time it takes for someone to test positive for COVID-19 and for their symptoms to worsen (for example, by escalating to hospitalization, which takes nearly a week [22] before resulting in death). For vaccination, a 14-day lag is taken, as immunity from vaccines is generally understood to develop two weeks or longer after receiving a shot [23–25].

The Oxford Containment and Health Index is calculated out of 100 where 100 means strict restrictions and 0 means no restrictions imposed on the general population. This variable is regressed with a 14-day lag, as we assume that any new government restrictions will take approximately that long to have any effect. Additionally, the time variable is meant to capture any unmeasured or seasonal effects on COVID-19 deaths in Pakistan, such as an overall rate of increase or decrease of daily deaths in each wave. We assume the error term is not correlated with any of the independent variables.

Newey-West standard errors are used to account for autocorrelation and potential heteroskedasticity in the error terms. Statistical tests are performed to ensure that the required assumptions for the regression model are met: for heteroskedasticity, the Breusch-Pagan test is applied, whereas for serial correlation, the Durbin-Watson test is used. Variance inflation factor (VIF) is calculated for multicollinearity. The presence of unit roots is tested using augmented Dicky-Fuller tests for each independent variable in our regression model. All the variables are found to be stationary, fulfilling an important pre-requisite for our analysis (Table 1). The statistical analysis is carried out using STATA 17 software.

## Results

### Summary statistics

Pakistan experienced five distinct waves from 3rd April 2020 till 23rd February 2022 (Fig 1). Wave 1 lasted the longest (150 days), while the wave 5 was the shortest (83 days). Wave 4 was remarkable for its relatively rapid upslope and a long tail, while wave 5 showed a reverse

**Table 1.  Augmented Dicky-Fuller tests for unit roots.**

| Variable | Test Statistics | P-Value | H₀: Random Walk |
|---|---|---|---|
| Daily New COVID-19 Deaths | -6.96 | 0.000 | Stationary |
| Daily New COVID-19 Cases | -3.23 | 0.018 | Stationary |
| Daily New COVID-19 Tests | -3.94 | 0.001 | Stationary |
| Oxford Containment and Health Index | -3.30 | 0.015 | Stationary |
| Ventilator-admitted Ratio | -5.69 | 0.000 | Stationary |
| Oxygen-Admitted Ratio | -3.00 | 0.034 | Stationary |
| Fully Vaccinated People | -2.77 | 0.003 | Stationary |

pattern. The duration of each wave of COVID-19 in Pakistan was shorter than the preceding one apart from wave 4. After wave 1, each wave took less time to reach its peak and took longer to reach its trough, apart from wave 5.

The capacity to conduct tests expanded over time from an average of 17,142 tests daily during wave 1 to 49,650 during wave 5. The increase in the daily tests peaked during wave 4. The highest average daily number of cases (3147) were observed during wave 3. The rate of increase of COVID-19 cases was the highest during wave 4, but the rate of decline in cases after the peak of a wave was the fastest during wave 5. Test-to-case ratio kept increasing from 15 during wave 1 to 57 during the wave 5. While total positivity varied across waves, the rate in daily change of positivity remained relatively unchanged apart from waves 2 and 3, where it was lower as compared to other waves (Table 2).

Hospitalizations were the highest for waves 1 and 3 and the lowest for wave 5, whereas duration of hospitalization fell linearly from an initial 13 days during wave 2 to 5 days during wave 5. Hospitalizations became more specific over time in that, nearly two thirds of admitted patients during wave 1 were stable, compared to 9% during wave 5. The average stable-admitted ratio decreased continuously from wave 1 to wave 4 but increased slightly in wave 5. The rate at which people recovered from COVID-19 and/or were discharged from hospital was the fastest in wave 4 but the slowest in wave 2.

The average oxygen beds-admitted ratio continuously increased in each wave, reaching its maximum value in wave 4. During wave 1, 27% of all admissions required oxygen and 7% needed a ventilator, compared to 81% and 10% respectively during wave 5. The average oxygen bed utilization followed a declining trend except for wave 3 (24%) and was the lowest in wave 5 (7%). The trend of average ventilators utilization ratio showed that all available ventilators were not fully utilized in any of the five waves. The highest ventilators utilization was in wave 3 (20%) and the lowest in wave 5 (5%). These two ratios suggest that most critical patients were put on oxygen for recovery and a small proportion of these people were transferred to ventilators.

Deaths from COVID-19 were the highest during wave 3 at 9,423, which also saw the highest daily number of deaths (78.5) and the highest rate of increase in daily deaths. Average daily deaths to hospitalization rate peaked during wave 2, while deaths to ventilator use was the highest during wave 1. Average deaths to case ratio was the highest for wave 3 but was in the 2.2–2.8% range, except for wave 5 when it was 1.1%.

Pakistan's vaccination drive started towards the end of wave 2, but full vaccination (i.e., people receiving both their doses) did not happen until the beginning of wave 3. Consequently, total and daily new second dose of vaccine administered was highest in wave 5. Government restrictions, measured by the Oxford Containment and Health Index, appeared to be comparable in each wave.

**Table 2. Key indicators and characteristics of each COVID-19 wave.**

| Variables | | Wave 1 | Wave 2 | Wave 3 | Wave 4 | Wave 5 |
|---|---|---|---|---|---|---|
| Dates | | April 3, 2020 to August 31, 2020 | October 12, 2020 to February 16, 2021 | February 23, 2021 to June 22, 2021 | July 6, 2021 to November 29, 2021 | December 7, 2021 to February 23, 2022 |
| **Main variants of concern (VOCs)** | | B.1 | B.1.36 | (Alpha/B.1.1.7)/ (Delta/B.1.617.2) | Delta/B.1.617.2 | Omicron/B.1.1.529 |
| **Wave timespan** | | | | | | |
| Duration of wave (days) | | 150 | 128 | 120 | 147 | 83 |
| Wave start till peak (days) | | 72 | 57 | 55 | 31 | 52 |
| Wave peak till trough (days) | | 78 | 71 | 65 | 116 | 31 |
| **COVID-19 tests** | | | | | | |
| Total tests (n) | | 2,571,244 | 4,638,219 | 5,458,041 | 7,148,130 | 4,120,909 |
| Avg. Daily new tests (n) | | 17,142 (7,931) | 36,236 (5,363) | 45,484 (8,623) | 48,627 (8,341) | 49,650 (7,494) |
| Δ Daily new tests | Till peak | 380.9 | 216.5 | 701.1 | 836.6 | 540.4 |
| Δ Daily new tests | After peak | -147.8 | -101.0 | -504.9 | -286.4 | -1137.6 |
| **COVID-19 cases** | | | | | | |
| Total cases (n) | | 293,752 | 246,188 | 377,668 | 320,333 | 221,825 |
| Avg. Daily new cases (n) | | 1,958 (1,687) | 1,923 (815) | 3,147 (1,547) | 2,179 (1,548) | 2,673 (2,519) |
| Δ Daily new cases | Till peak | 93.9 | 60.9 | 94.0 | 161.0 | 155.9 |
| Δ Daily new cases | After peak | -84.8 | -40.0 | -84.1 | -48.0 | -236.4 |
| **Test-to-case ratio** | | | | | | |
| Avg. Test-to-case ratio (%) | | 15 (15) | 23 (11) | 19 (11) | 44 (38) | 57 (56) |
| Δ Test-to-case | Till peak | 4.1 | -1.1 | -0.4 | -1.13 | -3.31 |
| Δ Test-to-case | After peak | 84.5 | 0.3 | 0.7 | 1.36 | 0.99 |
| **COVID-19 positivity rate** | | | | | | |
| Avg. Daily positivity rate (%) | | 11.3 (6.4) | 5.2 (1.8) | 6.8 (2.9) | 4.2 (2.6) | 4.9 (4.1) |
| Δ Positivity rate | Till peak | 0.23 | 0.15 | 0.10 | 0.23 | 0.22 |
| Δ Positivity rate | After peak | -0.28 | -0.10 | -0.11 | -0.07 | -0.30 |
| **Hospitalization and treatment** | | | | | | |
| Avg. hospital admissions at any given time (n) | | 3,817 (2,479) | 2,187 (818) | 3,970 (1,366) | 3,396 (1,776) | 1125 (423) |
| Avg. Length of hospital stay (days)[a] | | - | 13 | 11 | 9 | 5 |
| Avg. Daily new hospitalization-to-cases (%) | | - | 8.8 (2.4) | 12.4 (3.5) | 22.7 (10.7) | 15.7 (10.8) |
| Avg. Daily recoveries (n) | | 1,870 (2,587) | 1,746 (1,608) | 2,990 (1,454) | 2,272 (2,329) | 2,371 (3,797) |
| Δ Daily recoveries | Till peak | 23.4 | 14.3 | 69.9 | 202.4 | 28.76 |
| Δ Daily recoveries | After peak | -19.8 | -6.6 | -51.1 | -57.7 | -0.97 |
| Avg. Stable-admitted (%) | | 67 (27) | 20 (6) | 14 (5) | 7 (4) | 9 (4) |
| Avg. Patients on oxygen-admitted (%) | | 27 (23) | 68 (5) | 75 (5) | 83 (2) | 81 (3) |
| Avg. Oxygen beds in use at any given time (n) | | 865 (828) | 1,521 (608) | 2,995 (1,080) | 2,803 (1,470) | 918 (361) |
| Avg. Oxygen beds utilization (%) | | 18 (11) | 15 (6) | 24 (8) | 20 (10) | 7 (3) |
| Avg. Ventilators in use at any given time (n) | | 214 (163) | 249 (90) | 429 (145) | 336 (153) | 109 (36) |
| Avg. Ventilator-admission (%) | | 7 (4) | 11 (1) | 11 (0.6) | 10 (2) | 10 (1) |
| Avg. Ventilators utilization (%) | | 17 (10) | 13 (5) | 20 (7) | 15 (7) | 5 (2) |
| **COVID-19 deaths** | | | | | | |
| Total deaths (n) | | 6,204 | 5,537 | 9,423 | 6,287 | 1,401 |
| Avg. Daily new deaths (n) | | 41.4 (35.3) | 43.3 (23.0) | 78.5 (37.4) | 42.8 (30.6) | 16.9 (14.3) |

*(Continued)*

**Table 2.** (Continued)

| Variables | | Wave 1 | Wave 2 | Wave 3 | Wave 4 | Wave 5 |
|---|---|---|---|---|---|---|
| Δ Daily new deaths | Till peak | 1.1 | 0.5 | 2.0 | 1.17 | 0.45 |
| Δ Daily new deaths | After peak | -1.0 | 0.1 | -1.9 | -0.44 | -0.81 |
| Avg. Daily new deaths-daily new hospitalization (%) | | 11 (5) | 27 (13) | 23 (8) | 10 (4) | 7 (4) |
| Avg. Daily new deaths-ventilator in use (%) | | 23 (17) | 17 (6) | 18 (5) | 12 (5) | 13 (8) |
| Avg. Death-to-case ratio (%) | | 2.2 (0.8) | 2.3 (1.1) | 2.8 (1.2) | 2.2 (0.8) | 1.1 (1.0) |
| **COVID-19 vaccination**[b] | | | | | | |
| Total administered 2nd doses of vaccine | | - | - | 3,136,386 | 45,294,948 | 45,667,572 |
| Avg. Daily new 2nd doses of vaccine | | - | - | 26,356 (25,980) | 308,129 (172,896) | 550,212 (186,287) |
| **Policy environment** | | | | | | |
| Oxford containment and health index | | 58.8 (7.6) | 62.4 (4.0) | 65.7 (6.3) | 58.1 (9.4) | 58.6 (5.5) |

Note: Standard deviations are reported in parentheses. Δ refers to rate of change.

[a]Data for daily new COVID-19 hospitalizations was not available during wave 1.

[b]COVID-19 vaccines were not being administered during waves 1 and 2.

## OLS regression results

The linear OLS regression results for daily new COVID-19 deaths indicate that daily new COVID-19 cases were a statistically significant determinant for daily new deaths in all five waves at 95% CI; a one-percentage increase in COVID-19 cases caused a 0.46–0.69% increase in deaths across the five waves (Table 3).

The daily new COVID-19 tests and Oxford containment and health index, which records the presence of government NPIs and restrictions, were both found to be statistically significant determinants of COVID-19 deaths in wave 1. Increasing daily new tests by 1% reduced daily deaths by 0.65% (95% CI: 0.26, 1.04). An increase in Oxford Containment and Health Index by 1 point resulted in 0.03% reduction in daily new deaths (95% CI: -0.05, -0.005).

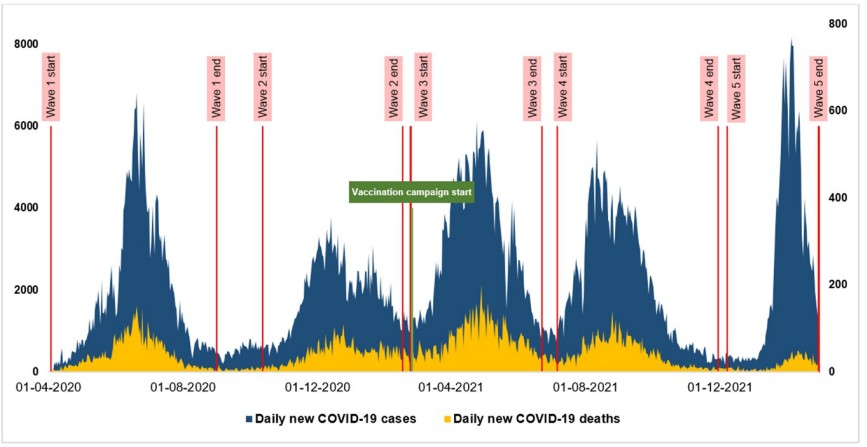

**Fig 1. COVID-19 waves, daily new cases, and daily new deaths.**

Table 3. OLS regression results showing associations of selected variables with daily new COVID-19 deaths.

| Dependent: Log daily new COVID-19 deaths ($Ln\ Y_t$) | (1) Wave 1 (N = 122) | (2) Wave 2 (N = 128) | (3) Wave 3 (N = 104) | (4) Wave 4 (N = 147) | (5) Wave 5 (N = 83) |
|---|---|---|---|---|---|
| Log daily new COVID-19 cases ($LnX1_{t+21}$) | 0.455* | 0.685* | 0.691* | 0.677* | 0.506* |
| | [0.289,0.620] | [0.430,0.939] | [0.358,1.025] | [0.542,0.812] | [0.189,0.823] |
| Log daily new COVID-19 tests ($LnX2_{t+28}$) | 0.648* | 0.308 | -0.162 | 0.268 | 0.296 |
| | [0.260,1.037] | [-0.179,0.795] | [-0.645,0.322] | [-0.122,0.657] | [-0.771,1.364] |
| Oxford containment and health index ($X3_{t+14}$) | -0.027* | -0.009 | 0.012 | 0.017 | -0.030 |
| | [-0.050, -0.005] | [-0.030,0.012] | [-0.005,0.028] | [-0.006,0.040] | [-0.072,0.013] |
| Time trend ($X4_t$) | -0.035* | -0.002 | -0.009* | -0.005* | 0.011 |
| | [-0.042, -0.028] | [-0.005,0.001] | [-0.016, -0.002] | [-0.010, -0.000] | [-0.008,0.030] |
| Ventilator-admitted ratio ($X5_t$) | -16.662* | 0.555 | -11.074 | -14.092* | -15.681* |
| | [-29.406, -3.918] | [-10.098,11.208] | [-25.332,3.183] | [-19.096, -9.089] | [-27.872, -3.489] |
| Oxygen-admitted ratio ($X6_t$) | 3.734* | 4.413* | 0.148 | 0.640 | 2.435 |
| | [0.788,6.680] | [1.848,6.979] | [-5.421,5.718] | [-2.558,3.838] | [-4.943,9.814] |
| Log second doses administered ($LnX7_{t+14}$) | - | - | 0.073* | 0.104* | -0.375* |
| | | | [0.006,0.139] | [0.010,0.198] | [-0.668, -0.083] |
| Constant | -1.222 | -6.707* | 3.169 | -3.417 | -4.846 |
| | [-4.180,1.736] | [-10.447, -2.967] | [-1.606,7.945] | [-8.348,1.514] | [-18.956,9.263] |

Note:

*Significant at 95% confidence interval. Newey-West confidence intervals in parentheses.

The time trend variable was statistically significant in waves 1, 3 and 4. The coefficients indicate that, on average, daily COVID-19 deaths decreased at a rate of 0.04% per day (95% CI: -0.04, -0.03) during wave 1. However, daily new deaths reduced at a rate of 0.01% per day during wave 3 (95% CI: -0.02, 0.002) and wave 4 (95% CI: -0.01, 0.00).

The ventilator-admitted ratio was statistically significant in waves 1, 4 and 5. The coefficient was negative throughout these three waves and significant at 95% CI. The coefficients indicates that if ratio of patients on ventilator out of the admitted increased then daily new deaths would decrease by 14–17%.

Oxygen-admitted ratio was only significant in waves 1 and 2 at 95% CI, where the coefficient was positive, implying that an increase in the ratio of oxygenated patients out of the total admitted increased was associated with an increase in daily new deaths due to COVID-19 by approximately 4%.

Lastly, the number of fully vaccinated people is statistically significant in each of the last three waves. During wave 3, COVID-19 deaths increased by 0.07% (95% CI: 0.006,0.14) as percentage of fully vaccinated people increased by one percent. This rate increased to 0.10% (95% CI: 0.006,0.14) during wave 4. However, during wave 5, daily new deaths due to COVID-19 decreased by 0.38% (95% CI: -0.67, -0.08) as fully vaccinated people increased.

## Discussion

Pakistan experienced 5 distinct waves from 3rd April 2020 to 23rd February 2022. Our analysis reflects both the evolution of Pakistan's response, as well as the differential impact of different variants of the virus shaped the contours and features of each wave. Pakistan experienced its initial wave earlier than other South Asian countries including India, Bangladesh, Sri Lanka and Nepal, while peaks for the subsequent waves coincided with those in other countries [26].

The upslope, as seen by the rate of change for testing and cases, was always steeper than during the downward slope of a wave. This pattern follows what is known about infectious epidemics in that cases rise quickly, plateau and then fall, slowly to an endemic state where a low ebb of infections persists in the community indefinitely [5]. In fact, each wave behaved as a typical epidemic caused by a distinct variant of the virus. Wave 1 was dominated by B.1 variant, wave 2 by B.1.36 variant, the wave 3 by Alpha/B1.1.7 and Delta/B.1.617.2 variants, wave 4 had majority cases of the Delta/B.1.617.2 variant [27–29] while wave 5 was driven by Omicron/BA.5.2.1.7 [4].

A key challenge faced by Pakistan at the beginning of the pandemic was that there was little prior experience with any pandemic outbreak of such level. Although disease surveillance systems exist, they had not been scaled to manage case surveillance, hospital admissions, daily deaths, and eventually large-scale adult vaccination and event tracking. Pakistan has a federal system of governance where provinces provide health services while the federal ministry provides guidance and coordination. In addition, considerable curative care is in the private sector. To address the potential difficulties in mounting a unified national response to the disease in the face of this diversity, a National Action Plan for COVID-19 was formulated in March 2020 that placed the responsibility for the national response in a National Coordination Committee (NCC) that was headed by the Prime Minister and attended by all federal ministers. The NCC set national policy which was implemented by the National Command and Operation Centre (NCOC) that was co-headed by the military and civilian leadership [30]. The NCOC coordinated the management of the extensive lockdowns, other key NPIs such as school closures, limited opening hours for essential businesses (examples of which included grocery stores and pharmacies), closure of borders, cancellation of public events and social gatherings [31,32]. This was supported based on an elaborate data gathering and analysis system that guided daily decisions.

Wave 1 continued the longest and intervals became shorter between each successive wave. Each wave showed unique features, that were determined by the particular variant that drove that wave, along with the larger context that included the type of the variant driving the wave, the extent and type of preventive interventions and eventually the availability of the vaccine.

Pakistan's response to COVID-19 evolved over time. For example, wave 1 had the highest positivity rates and the longest duration, in part due to low initial rates of testing, including very little contact tracing in the early days [33]. As testing increased and mobility restrictions tightened, duration of waves 2 and 3 became shorter. However, by the end of wave 2, intervention fatigue had set in. Implementation was laxer, and these factors contributed to more cases and deaths of any wave during wave 3. Indeed, the Oxford Containment and Health Index was significant only during wave 1 in terms of preventing deaths.

In addition to preventive measures, the higher daily COVID-19 cases in waves 3, 4, and 5 may be attributed to highly transmissible Alpha [34,35], Delta [36,37] and Omicron [38] variants, and to easing of severe restrictions such as lockdowns and school closures [39]. It is also possible that many cases were missed during wave 1 due to limited testing. However, the stability of daily testing in waves 3 to 5 suggests a stable equilibrium between the testing system and how cases were being incident–the system was capturing most of the cases from previously recognized populations and locations. It is likely that undiagnosed cases and deaths were few, since as part of the national surveillance, teams kept abreast of burials in large and midsized towns and also periodically canvased opinion of general practitioners about upsurges in respiratory illnesses. On average, Pakistan had fewer cases per million population than neighboring countries of India, Bangladesh, and Iran, as well as several of the developed countries [26].

As with prevention, clinical management of cases evolved over time. Initially most cases were hospitalized as seen by the high case to hospitalization ratio–only 27% of admissions

required oxygen 7% required ventilators during wave 1. In fact, there was a correlation between deaths and oxygenation (which was mostly at hospitals) during waves 1 and 2, a pattern that was seen globally. However, with each succeeding wave, use of oxygen increased while ventilators fluctuated within a narrow range, as was also seen in India [40,41]. Thus, even as COVID-19 hospitalizations peaked during wave 3, hospitalization to case ratio increased, and average duration of hospitalization and the use of hospitals for simple oxygenation fell, suggesting hospitals, ICU and ventilators, were increasingly reserved only for the sickest [42]. Deaths correlated best with a 21-day delay model rather than a 28-day one, suggesting that most deaths happened early after infection. Higher hospitalizations during wave 3 may also have been attributed to the Alpha followed by Delta variants [43–45]. By contrast, lower hospitalizations, length of stay, and mortality during wave 5 may be attributed to the Omicron variant that was seen worldwide [46,47], and specifically in South Africa [48] and Brazil [49] during the Omicron waves. Vaccination started earlier on in wave 3 and more than half of the eligible population was fully vaccinated by wave 5 [3] and may have contributed to lower hospitalizations in wave 5. Unlike COVID-19 induced major challenges to the healthcare capacity in various countries [50,51], Pakistan was able to build healthcare resources capacity to keep pace with the pandemic. Ventilator and oxygen utilization never exceeded 20% and 24% respectively in wave 3.

Vaccination drive started in Pakistan by the end of February 2021. Despite a slow start, vaccination picked up pace from 26,356 daily vaccinations in wave 3 to 308,129 in wave 4 as it was rolled-out to younger population and vaccine supply increased in the country. Average daily deaths did not reduce significantly due to vaccinations during waves 3 and 4 [52,53], but showed marked reduction in hospitalizations and deaths towards the end of wave 4 and during the entire wave 5 [54].

From our regression model, we found that daily new COVID-19 cases were statistically significant determinants of daily new deaths due to COVID-19. The association was also observed from the wave 3 as both cumulative cases and deaths were the highest, including the average daily deaths which were considerably higher than any other wave, as seen in other countries [55]. Secondly, daily new deaths due to COVID-19 increased with patients on oxygenated beds while decreased with patients on ventilators in the initial waves, potentially due to high patients load in hospitals, critical patients were put on oxygen rather than ventilator. Wave 5 experienced the smallest number of daily COVID-19 deaths possibly because it was dominated by the Omicron variant [56].

## Limitations

There are several limitations associated with the data used in this paper. While the official data used for the analysis are disaggregated by sub-national level, demographic disaggregation, such as age or gender, are not available. This limits the analysis in terms of the implication of gender and age on COVID-19 deaths. The national data is compiled by aggregating the numbers for each subnational unit in Pakistan. However, such an analysis would be too extensive to depict and therefore our analysis does not account for subnational differences. It is possible that distinctive cultures, behaviors, and differences in the stringency in enforcement of interventions vary between regions and may in theory, influence the number of COVID-19 cases and deaths.

Similarly, data for hospitalizations is also unaccompanied by any information on comorbidities, as this information was not available beyond treating hospitals, losing a level of richness of analysis that includes such comorbidities. Also, data for daily new hospital admissions

started becoming available towards the very end of wave 1. Consequently, the average length of hospital stay could not be calculated for this wave.

The official vaccination data available to us at the time of this analysis is not desegregated by the different types of available vaccines, for example Sinopharm, CanSino, Sputnik V and others. Differential impact of each vaccine on COVID-19 deaths in Pakistan would be informative. All the above limitations notwithstanding, we are confident that this study provides crucial insights into the prevailing trends of COVID-19 in Pakistan in manner that is constructive.

## Conclusion

We describe how COVID-19 waves differed in terms of cases, hospitalizations, and deaths in Pakistan, and analyze potential reasons for these differences. Pakistan experienced its initial COVID-19 wave earlier than other South Asian countries, with wave 1 lasting the longest. As testing increased and restrictions were enforced, subsequent waves became shorter, but wave 3 stood out due to lax implementation, resulting in the highest number of cases and deaths. The higher daily cases in waves 3, 4, and 5 were also attributed to the highly infectious Delta and Omicron variants. Wave 3 recorded the most COVID-19 deaths, with 9,423 fatalities, the highest daily death rate, and the steepest increase in daily deaths. Lastly, vaccination began in wave 2, with full vaccination achieved in wave 3, and the highest second-dose vaccinations occurred in wave 5.

At the pandemic's onset, Pakistan's lack of prior experience was a challenge. However, a National Action Plan for COVID-19 was established in March 2020. COVID-19 management in Pakistan kept pace with the spread of the disease during five distinct waves and successfully implemented the COVID-19 vaccination drive nationwide. The experiences and limitations offer valuable insights for future pandemic management for a developing country like Pakistan.

## Supporting information

**S1 Table. Calculated ratio variables and their descriptions.**
(DOCX)

## Author Contributions

**Conceptualization:** Taimoor Ahmad, Mujahid Abdullah, Abdul Mueed, Faisal Sultan, Ayesha Khan, Adnan Ahmad Khan.

**Data curation:** Mujahid Abdullah, Abdul Mueed, Faisal Sultan.

**Formal analysis:** Taimoor Ahmad, Mujahid Abdullah.

**Funding acquisition:** Adnan Ahmad Khan.

**Investigation:** Taimoor Ahmad, Abdul Mueed, Ayesha Khan, Adnan Ahmad Khan.

**Methodology:** Taimoor Ahmad, Mujahid Abdullah, Abdul Mueed.

**Project administration:** Ayesha Khan, Adnan Ahmad Khan.

**Software:** Taimoor Ahmad, Mujahid Abdullah.

**Supervision:** Faisal Sultan, Ayesha Khan, Adnan Ahmad Khan.

**Validation:** Taimoor Ahmad, Mujahid Abdullah, Abdul Mueed, Faisal Sultan, Ayesha Khan, Adnan Ahmad Khan.

**Visualization:** Faisal Sultan.

**Writing – original draft:** Taimoor Ahmad, Mujahid Abdullah, Abdul Mueed.

**Writing – review & editing:** Faisal Sultan, Ayesha Khan, Adnan Ahmad Khan.

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
