## [Decision Letter · Decision Letter 0]

8 Sep 2023

PONE-D-23-01793COVID-19 in Pakistan: A national analysis of five pandemic wavesPLOS ONE

Dear Dr. Khan,

Thank you for submitting your manuscript to PLOS ONE. After careful consideration, we feel that it has merit but does not fully meet PLOS ONE’s publication criteria as it currently stands. Therefore, we invite you to submit a revised version of the manuscript that addresses the points raised during the review process.

ACADEMIC EDITOR: Please address the reviewers' comments, especially their major concerns. In particular, your statement that seems to imply that vaccinations led to higher mortality needs to be changed. The risk of confounding bias in such study types needs to be highlighted which may have led to spurious findings which can occur even after adjustment due to residual confounding. ==============================

We look forward to receiving your revised manuscript.

Kind regards,

Huzaifa Ahmad Cheema

Academic Editor

PLOS ONE

Journal Requirements:

“This work was supported, in whole or in part, by the Bill & Melinda Gates Foundation [grant number: INV-025171]. Under the grant conditions of the Foundation, a Creative Commons Attribution 4.0 Generic License has already been assigned to the Author Accepted Manuscript version that might arise from this submission. The funders had no role in study design, data collection and analysis, decision to publish, or preparation of the manuscript.”

Reviewers' comments:

Reviewer's Responses to Questions

**Comments to the Author**

1. Is the manuscript technically sound, and do the data support the conclusions?

Reviewer #1: Yes

Reviewer #2: Partly

Reviewer #3: Yes

Reviewer #4: Partly

2. Has the statistical analysis been performed appropriately and rigorously? 

Reviewer #1: Yes

Reviewer #2: Yes

Reviewer #3: Yes

Reviewer #4: Yes

3. Have the authors made all data underlying the findings in their manuscript fully available?

Reviewer #1: Yes

Reviewer #2: Yes

Reviewer #3: Yes

Reviewer #4: Yes

4. Is the manuscript presented in an intelligible fashion and written in standard English?

Reviewer #1: Yes

Reviewer #2: Yes

Reviewer #3: Yes

Reviewer #4: Yes

5. Review Comments to the Author

Reviewer #1: The manuscript is a nation-wide review of the first five waves of COVID-19 in Pakistan. Although much has been written about COVID-19, nation-wide data from an under-resourced setting are lacking, thus this manuscript is a valuable additional to the medical literature. It offers a fairly comprehensive review of the epidemiology of these waves in Pakistan, covering a number of important and interesting variables. The manuscript is well written, completed an appropriate analysis, and is well-cited. Although I rarely make this determination in reviewing manuscripts for the first time, I believe it is ready to accept for publication as it currently stands. My congratulations to the authors on their excellent work.

Reviewer #2: This is an interesting manuscript on a timely subject, which brings valuable information and evidence on the COVID-19 pandemic.

The main observation concerns the objectives of this analysis: they should be clearly and unambiguously stated. The conclusions should state whether (and how) they were met.

Major observations:

(1) Page 6, line 128: the manuscript comprises statistical analysis and inferences.

(2) Page 5 and page 7 -- lists of themes and independent variables in the regression model, respectively: the presumed connection between the two lists needs to be revised and clarified; if there is no connection, the lists should nevertheless be revised for more clarity and meaningfulness (namely, remove all ambiguity).

(3) Page 8, line 162: Statistical(!) tests were performed to insure that the required assumptions for the regression model were met.

When revising, please also consider the following:

(A) Page 5, line 93: double-check the reference to Table 1.

(B) Page 6, Model specification: the model itself should be independent of the statistical package employed to analyze the data; I would kindly suggest specifying the statistical package at the end of the section.

(C) Page 6, line 132 (equation): t must be detailed/explained; equation should also have a number.

(D) Pages 13-14, Table 3: the outcome/dependent variable(s) must be clearly stated, together with the number of data records; a landscape layout should be considered, as well. I also recommend having a more informative caption: the explicit/unambiguous connection with the regression model.

Additional comments:

(a) Page 3 – Introduction: the citations and references' numbering is incorrect; please revise.

(b) Table S1 – length of hospital stay: the terms "hospital admissions" and "new hospital admissions" are confusing. "Hospital admissions" seems to refer to the number of hospitalized patients at a given time. Please revise the definition and remove any ambiguity.

(c) Page 3, line 51: please revise the SARS-CoV-2 strains, namely the Alpha strain notation) and also revise the tables that include the notation.

(d) Page 8, Table 3: the connection with the regression model's predictors should be explicit (for example by adding an additional column).

(e) Pages 9-12, Table 2: the table is very long and confusing due to the lack of horizontal lines. Authors might consider splitting into meaningful separate tables with landscape layout.

(f) Authors should use a consistent notation of the pandemic waves throughout the manuscript (for example, choose one of "wave 3" or "third wave").

(g) Please double-check all specifications of confidence intervals throughout the manuscript (an example of faulty notation is on page 13, line 225).

Reviewer #3: Well scientific written manuscript. Need to address following few questions

1. describe little bit about specific criteria in abstract method

2. The rationale and justification for this article must be incorporated

3. Please mention impact of study

4. Provincial specific cultural and social, religious, education status, social mobalization, public compliance, administrative, infrastructural (etc) factors must be considered for NPI. Did these factors may have played a role as confounder.

Reviewer #4: The article under review provides a comprehensive overview of the key parameters associated with five waves of COVID-19 in Pakistan. The paper presents pertinent information and employs statistical methodology. While the manuscript is well-structured overall, I have a few points of feedback to share.

1. Regarding the statement in the summary, "A one percent increase in vaccinations increased daily new COVID-19 deaths by 0.10% (95% CI: 0.01, 0.20)," it is essential to clarify the source of this data, since it is not clear in the text or in the tables. The current phrasing might inadvertently imply a causal relationship between vaccination and an increase in daily deaths. It is crucial to revise this sentence to avoid the misunderstanding that vaccination directly led to higher mortality. Although the data allows for different interpretations, a clear causal link cannot be established solely from these numbers.

2. Consider incorporating a comparative analysis between the official death data examined in your study and the excess mortality data. This comparison is pertinent since the cumulative death count over the five waves appears to be considerably lower than the excess mortality figures reported in early 2022 (https://doi.org/10.1016/S0140-6736(21)02796-3). Additionally, the insights provided in this article could benefit from a cross-reference to related research, as indicated by this source: https://doi.org/10.1038/s41586-022-05522-2.

3. Enhance the clarity of the visual representation by marking the initiation and conclusion of each wave within the figure 1. To provide a more comprehensive perspective, it would be valuable to disaggregate the data by gender, including both cases and fatalities. Furthermore, it could be beneficial to highlight the time points at which vaccination initiatives were introduced.

4. Concerning Table 3, the significance of the asterisk accompanying all values of LOG (Daily New COVID-19 Cases) – 21-day delay) requires clarification. Please elucidate the purpose of these asterisks to aid the reader's understanding of the presented data.

6. PLOS authors have the option to publish the peer review history of their article (what does this mean?). If published, this will include your full peer review and any attached files.

Reviewer #1: No

Reviewer #2: No

Reviewer #3: **Yes: **Ehsan Ahmed Larik

Reviewer #4: No

---

## [Author Response · Author response to Decision Letter 0]

30 Oct 2023

Reviewer #1: 

The manuscript is a nation-wide review of the first five waves of COVID-19 in Pakistan. Although much has been written about COVID-19, nation-wide data from an under-resourced setting are lacking, thus this manuscript is a valuable additional to the medical literature. It offers a fairly comprehensive review of the epidemiology of these waves in Pakistan, covering a number of important and interesting variables. The manuscript is well written, completed an appropriate analysis, and is well-cited. Although I rarely make this determination in reviewing manuscripts for the first time, I believe it is ready to accept for publication as it currently stands. My congratulations to the authors on their excellent work. 

Comment : We are very grateful for your review and comments on our manuscript. We are happy to contribute to the COVID-19 literature, especially for a country like Pakistan.

Reviewer #2: 

The main observation concerns the objectives of this analysis: they should be clearly and unambiguously stated. The conclusions should state whether (and how) they were met. 

Comment #1: We have improved the objective of the manuscript in the last paragraph of introduction. Lines 80-87

Comment #2: We have also updated the conclusion to state how the objectives were met. Lines 374-390

Page 6, line 128: the manuscript comprises statistical analysis and inferences. 

Comment: We have improved the wording. Line 132 -133

Page 5 and page 7 -- lists of themes and independent variables in the regression model, respectively: the presumed connection between the two lists needs to be revised and clarified; if there is no connection, the lists should nevertheless be revised for more clarity and meaningfulness (namely, remove all ambiguity). 

Comment: We have made some changes in the data and variables section to distinguish between the themes and independent variables in the regression model. 

Page 8, line 162: Statistical(!) tests were performed to insure that the required assumptions for the regression model were met. 

Comment: We have replaced the wording of this line as suggested. Line 166-169

Page 5, line 93: double-check the reference to Table 1. 

Comment: The reference is updated to table 2 which presents the length of each wave in days with their respective dates. 

Line 97 Page 6, Model specification: the model itself should be independent of the statistical package employed to analyze the data; I would kindly suggest specifying the statistical package at the end of the section. 

Comment: We have moved the statistical package information to the end of the section. Line 172-173.

Page 6, line 132 (equation): t must be detailed/explained; equation should also have a number. 

Comment: We have assigned equation number and explanation of t is provided. Line 128-145

Pages 13-14, Table 3: the outcome/dependent variable(s) must be clearly stated, together with the number of data records; a landscape layout should be considered, as well. I also recommend having a more informative caption: the explicit/unambiguous connection with the regression model. 

Comment: We have clearly stated the dependent variable, added data records in the headings and also updated the caption for more clarity. Line 230-231

Page 3 – Introduction: the citations and references' numbering is incorrect; please revise. 

Comment: We have revised and corrected the references in our entire manuscript.

Table S1 – length of hospital stay: the terms "hospital admissions" and "new hospital admissions “are confusing. "Hospital admissions" seems to refer to the number of hospitalized patients at a given time. Please revise the definition and remove any ambiguity. 

Comment: We have improved the wording of the definition to make it more clear. 

Page 3, line 51: please revise the SARS-CoV-2 strains, namely the Alpha strain notation) and also revise the tables that include the notation.

Comment: We have revised the SARS-COV-2 strains name both in line 54-55 and table 2. 

Page 8, Table 3: the connection with the regression model's predictors should be explicit (for example by adding an additional column).

Comment: Along with specifying the dependent variable in Table 3, we have included variable names as previously specified in equation 1 in Model Specification section to improve clarification.

Pages 9-12, Table 2: the table is very long and confusing due to the lack of horizontal lines. Authors might consider splitting into meaningful separate tables with landscape layout. 

Comment: We have added lines to separate each of the broad themes of indicators, shortened variable names, and revised for consistency to improve readability. Now Table 2 is set on a single page.

Authors should use a consistent notation of the pandemic waves throughout the manuscript (for

example, choose one of "wave 3" or "third wave"). 

Comment: We have made the notation of the pandemic waves consistent throughout the manuscript now.

Please double-check all specifications of confidence intervals throughout the manuscript (an example of faulty notation is on page 13, line 225). 

Comment: We have checked all the specification of confidence interval throughout the manuscript. We have removed the percentage sign from the confidence intervals. 

Reviewer #3: 

describe little bit about specific criteria in abstract method 

Comment: We have described the criteria for a wave in the methods section in abstract

The rationale and justification for this article must be incorporated 

Comment: We have made changes to the last paragraph of the introduction section to state our rationale and justification for this article clearly. 

Please mention impact of study 

Comment: Impact of the study is elaborated in the conclusion section of this manuscript.

Provincial specific cultural and social, religious, education status, social mobilization, public compliance, administrative, infrastructural (etc) factors must be considered for NPI. Did these factors may have played a role as confounder. 

Comment: The manuscript is primarily based on the national rather than provincial level. In addition, uniform NPIs were enforced across the country regardless of the province like school closure to reduce the spread of COVID-19 in the general population. 

Reviewer #4: 

Regarding the statement in the summary, "A one percent increase in vaccinations increased daily new COVID-19 deaths by 0.10% (95% CI: 0.01, 0.20)," it is essential to clarify the source of this data, since it is not clear in the text or in the tables. The current phrasing might inadvertently imply a causal relationship between vaccination and an increase in daily deaths. It is crucial to revise this sentence to avoid the misunderstanding that vaccination directly led to higher mortality. Although the data allows for different interpretations, a clear causal link cannot be established solely from these numbers. 

Comment: We have specified the source of data in the methods section of the abstract. We have also changed the wording of the vaccination variable interpretation. We have specifically used the term association now so readers do not consider it as a causal relationship.

Consider incorporating a comparative analysis between the official death data examined in your study and the excess mortality data. This comparison is pertinent since the cumulative death count over the five waves appears to be considerably lower than the excess mortality figures reported in early 2022 (https://doi.org/10.1016/S0140-6736(21)02796-3). Additionally, the insights provided in this article could benefit from a cross-reference to related research, as indicated by this source: https://doi.org/10.1038/s41586-022-05522-2. 

Comment: The excess mortality figures are derived from statistical modeling, and they estimate the total number of deaths in Pakistan, including those that may not have been officially reported. However, our data source for Pakistan is based solely on the figures reported by the Government of Pakistan. Comparing these reported statistics with the estimated total deaths due to COVID-19 may not yield meaningful or logically consistent results, as we would be essentially comparing confirmed cases with a broader estimate that includes unreported cases. This is, however, an interesting comparison which we may consider for our future research.

Enhance the clarity of the visual representation by marking the initiation and conclusion of each wave within figure 1. To provide a more comprehensive perspective, it would be valuable to disaggregate the data by gender, including both cases and fatalities. Furthermore, it could be beneficial to highlight the time points at which vaccination initiatives were introduced. 

Comment: Unfortunately, we have national aggregated data so we cannot separate it by gender, age etc. However, we’ve included daily new deaths due to COVID-19 and marked start and end of waves, as well as the starting date of vaccination campaign. 

Concerning Table 3, the significance of the asterisk accompanying all values of LOG (Daily NewCOVID-19 Cases) – 21-day delay) requires clarification. Please elucidate the purpose of these asterisks to aid the reader's understanding of the presented data. 

Comment: The asterisks represents the level of significance at 95% or more confidence level. This is already reported under the table 3.

---

## [Decision Letter · Decision Letter 1]

14 Dec 2023

COVID-19 in Pakistan: A national analysis of five pandemic waves

PONE-D-23-01793R1

Dear Dr. Khan,

We’re pleased to inform you that your manuscript has been judged scientifically suitable for publication and will be formally accepted for publication once it meets all outstanding technical requirements.

Kind regards,

Huzaifa Ahmad Cheema

Academic Editor

PLOS ONE

Additional Editor Comments (optional):

The authors have satisfactorily addressed the revisions.

Reviewers' comments:

Reviewer's Responses to Questions

**Comments to the Author**

1. If the authors have adequately addressed your comments raised in a previous round of review and you feel that this manuscript is now acceptable for publication, you may indicate that here to bypass the “Comments to the Author” section, enter your conflict of interest statement in the “Confidential to Editor” section, and submit your "Accept" recommendation.

Reviewer #2: All comments have been addressed

Reviewer #3: All comments have been addressed

Reviewer #4: All comments have been addressed

2. Is the manuscript technically sound, and do the data support the conclusions?

Reviewer #2: Yes

Reviewer #3: Yes

Reviewer #4: Yes

3. Has the statistical analysis been performed appropriately and rigorously? 

Reviewer #2: Yes

Reviewer #3: Yes

Reviewer #4: Yes

4. Have the authors made all data underlying the findings in their manuscript fully available?

Reviewer #2: Yes

Reviewer #3: Yes

Reviewer #4: Yes

5. Is the manuscript presented in an intelligible fashion and written in standard English?

Reviewer #2: Yes

Reviewer #3: Yes

Reviewer #4: Yes

6. Review Comments to the Author

Reviewer #2: The Authors have carefully addressed all the issues raised by the reviewers and the manuscript has been substantially improved. It will certainly convey the intended comprehensive analysis and will also bring a valuable insight into the COVID-19 related mortality in Pakistan, a country from the South Asian region for which there is still little evidence regarding the pandemic.

I would kindly make two recommendations to further improve the quality:

(a) a professional scientific proofreading would help addressing the few language issues still present in the manuscript;

(b) to improve the readability of Table 2 (page 10), Authors might consider either adding % when numbers in parentheses are percentages (rather than standard deviation values) or expressing the statistics related to the numerical variables as mean(i.e., average) +/- standard deviation.

Reviewer #3: All comments have been addressed. A much better revised article submitted. No specific advise. The article is clear, correct, and unambiguous.

Reviewer #4: The authors have implemented the majority of the suggested revisions provided by the reviewers. The manuscript has improved clarity.

7. PLOS authors have the option to publish the peer review history of their article (what does this mean?). If published, this will include your full peer review and any attached files.

Reviewer #2: No

Reviewer #3: **Yes: **Dr. Ehsan Ahmed Larik

Reviewer #4: No

---

## [Editor Report · Acceptance letter]

18 Dec 2023

PONE-D-23-01793R1 

PLOS ONE

Dear Dr. Khan, 

I'm pleased to inform you that your manuscript has been deemed suitable for publication in PLOS ONE. Congratulations! Your manuscript is now being handed over to our production team.

Kind regards, 

on behalf of

Dr. Huzaifa Ahmad Cheema 

Academic Editor

PLOS ONE